# Effect of semen dilution rate and dimethyl acetamide levels on post-thaw motility and fertility parameters of rooster sperm

Mohamed Shawky[1], Ahmed F. Fawy[2], Ahmed M. Elomda[3], Mohamed A. Elmenawey[2], Abd El-Rahman M. Atta[2], Ahmed O. Abbas[4], Gamal M. K. Mehaisen[2]*

1 Avian Research Center, King Faisal University, Al-Ahsa, Saudi Arabia, 2 Department of Animal Production, Faculty of Agriculture, Cairo University, Giza, Egypt, 3 Department of Animal Biotechnology, Animal Production Research Institute, Agriculture Research Center, Dokki, Giza, Egypt, 4 Department of Animal and Fish Production, College of Agricultural and Food Sciences, King Faisal University, Al-Ahsa, Saudi Arabia

* gamoka@cu.edu.eg

## Abstract

This study evaluated the impact of pre-freezing semen dilution rate and dimethyl acetamide (DMA) concentration on the post-thaw motility and fertility of cryopreserved rooster sperm. Rooster ejaculates were diluted with a standard EK extender to achieve low (LSC; $1 \times 10^9$ sperm/mL) and high (HSC; $2 \times 10^9$ sperm/mL) sperm concentrations. Each dilution group was further treated with three DMA concentrations (3%, 6%, or 9%) before cryopreservation. Post-thaw sperm motility traits were obtained by computer-assisted sperm analysis (CASA), and fertility features were evaluated through artificial insemination in hens. The current results showed that HSC significantly improved total motility (TM), curvilinear velocity (VCL), amplitude of lateral head displacement (ALH), and beat cross frequency (BCF), but reduced linearity (LIN) and straightness (STR) compared to LSC. DMA concentration had a quadratic effect on motility, with 6% yielding the highest progressive motility (PM), straight line velocity (VSL), and BCF. Fertility outcomes revealed that HSC resulted in higher fertilization rates, while neither DMA concentrations nor their interaction with dilution rates exerted significant effects on fertility traits. VCL, ALH, and BCF showed positive correlations with pipping-chicks rates, whereas STR, LIN, and WOB displayed negative correlations. These findings underscore the critical interplay between dilution rate and cryoprotectant concentration and provide practical guidance for developing more reliable cryopreservation protocols that can be applied under field conditions to enhance fertility management in poultry production.

## Introduction

Sperm cryopreservation remains the most applicable approach for facilitating artificial insemination, disease control, breeding, and selection programs in poultry species [1].

**Data availability statement:** All relevant data are within the manuscript and its Supporting Information files.

**Funding:** This research was financially supported by internal funds from the Academy of Scientific Research and Technology, Egypt (ASRT5105) and external funds from the Vice Presidency for Graduate Studies and Scientific Research at King Faisal University, Saudi Arabia (Project KFU252322). There was no additional external funding received for this study. The funders had no role in study design, data collection and analysis, decision to publish, or preparation of the manuscript.

**Competing interests:** The authors have declared that no competing interests exist.

The process of semen cryopreservation includes several intricate and essential steps that interact with each other and affect sperm capability post-thawing. Among these steps are those associated with the source and processing of semen before freezing, such as male donor, ejaculate collection, manipulation time, and extender composition and dilution rate, and those associated with the freezing technique, such as cryoprotectant (CPA) concentration, CPA equilibration, packaging type, freezing rate, CPA removal (if glycerol is utilized), and thawing procedure [2,3]. However, the previous studies have shown that frozen/thawed semen in the poultry industry remains restricted and not economically feasible because avian spermatozoa exhibit high levels of sensitivity, damage, and infertility rates after undergoing cryopreservation procedures [4–8].

Dimethyl acetamide (DMA), consisting of one amide group connected to two methyl groups [$CH_3$-C(=O)-N($CH_3$)$_2$], is commonly used as an intracellular cryoprotectant for semen freezing in poultry species [9,10]. It is suggested that DMA can penetrate cell membranes rapidly and participate in the natural biological processes of the cells [11]. DMA freezing technology is based on its capacity to substitute the intracellular water of sperm cells, thereby mitigating the harms caused by ice formation [12]. In contrast, the initial interaction of sperm with DMA during the freezing/thawing process may influence the membrane fluidity and acrosome reaction, probably exerting adverse effects on post-thaw sperm viability [13,14]. Earlier studies implemented DMA with semen extenders at wide levels ranging from 3–26% to boost the effectiveness of chicken sperm cryopreservation [4–6,15–21]. However, it was found that an appropriate sperm motility across various poultry species was obtained when applying semen freezing with a low concentration of DMA not exceeding 3–6% [5,6,17,18]. In contrast, the average fertility levels obtained from cryopreserved chicken semen utilizing DMA remain low and exhibit considerable variability, ranging from 10% to 77% [6,19,20,22].

The dilution of semen is a valuable approach for evaluating and/or increasing the fertilizing capability and hatchability of individual broiler breeder males [23,24]. It is well known that semen dilution may affect the viscosity, in turn, affecting sperm motility [25]. Standardizing semen dilution at the initiation stage of cryopreservation protocols is essential because of the individual variation in ejaculated sperm concentration from the birds [26]. Studies revealed that pre-freezing sperm concentration influenced the cryosurvival and quality of post-thaw sperm cells in turkey [27]. According to Zaniboni et al. [18], sperm concentration in rooster semen affected sperm motility and viability during semen processing. They reported that high sperm dilution to $1 \times 10^9$ sperm/mL had a beneficial impact on sperm motility. Due to the variation in the amount of sperm cells per millilitre of seminal plasma and extender, Iaffaldano et al. [28] discovered that a dilution rate of 1:4 was more successful than 1:3 in turkey semen. They also ascribed this impact to variations in the levels of DMA in each sperm cell or in the amounts of seminal plasma in the sperm sample's overall volume. Furthermore, seminal plasma appears to have a detrimental impact on sperm cells during both liquid storage [29] and cryopreservation [28]. It has been suggested that extracting seminal plasma from turkey spermatozoa prior to semen freezing might increase reproduction rates [30].

To our knowledge, no previous studies have conclusively determined the optimal dilution rate for chicken semen prior to freezing, particularly in relation to using various levels of dimethyl acetamide as a cryoprotectant. While research has independently examined the influence of DMA concentration and sperm dilution on cryosurvival in poultry species, the interactive effects of these two critical factors on rooster semen cryopreservation remain largely unexplored. This study, therefore, addresses this knowledge gap by evaluating how different pre-freezing semen dilution rates, in combination with varying DMA concentrations, influence the post-thaw motility and fertilizing capacity of rooster sperm. By integrating these two determinants, the present work provides novel insights into refining cryopreservation protocols aimed at enhancing both sperm survival and fertility outcomes in poultry.

## Materials and methods

### Ethical statement

This experimental study was conducted at the Agricultural Experiments Station (Cairo University, Egypt). This study did not involve animal suffering or sacrifice. All experimental protocols were authorized and approved by the Cairo University-Institutional Animal Care and Use Committee (CU-II-F-12-20).

### Management of birds

A total of sixteen roosters (55-wk-old) and eighty hens (40-wk-old) of Cairo-B2 chicken strain [5] were employed for the current study. The birds were placed in a semi-open house in individual battery cages (50 × 50×60 cm) and fed on a standard commercial breeder ration containing 14% crude protein, 2750 Kcal metabolizable energy, 4.15% crude fiber, and 3.3% crude fat (Feedmix-Egypt Co., Obour city, Kaliobeya, Egypt). All birds were provided with *ad libitum* access to feed and water and maintained at a 16-hour light cycle throughout the experimental period.

### Experimental design

Semen was ejaculated from the roosters twice a week by a lateral pressing of the cloaca after a few seconds of dorso-abdominal massage, as Bakst and Dymond [31] outlined. Semen ejaculates of the first two weeks were neglected, and then the semen ejaculates for 4 consecutive weeks were received in sterilized glass tubes. The individual ejaculates were selected subjectively based on the high gross motility and concentration to be pooled. An objective evaluation based on sperm concentration and progressive motility was then performed on each pool, as mentioned later. The semen pools with at least $4 \times 10^9$ sperm/mL and 60% progressive motility were divided into two groups according to the dilution rate with EK extender at low (LSC) and high (HSC) sperm concentration. The LSC and HSC groups were diluted with a standard EK semen extender until reaching $1 \times 10^9$ and $2 \times 10^9$ sperm/mL, respectively. The initial temperature at the time of dilution was 25°C. The EK extender for semen dilution consisted of 1.4 g of sodium glutamate (03956, Loba Chemie Pvt. Ltd., Colaba,

Mumbai, India), 0.98 g of disodium hydrogen phosphate (05971, Loba), 0.7 g of glucose (49139, Sigma-Aldrich, Inc., St. Louis, MO, USA), 0.7 g of inositol (04192, Loba), 0.21 g of sodium di-hydrogen phosphate (5859A, Loba), 0.2 g of D-fructose (03880, Loba), 0.14 g of potassium citrate (IP006, Loba), 0.1 g of polyvinylpyrrolidone (5315D, Loba), and 0.02 g of protamine sulfate (P3369, Sigma), all dissolved in 100 ml of bi-distilled water (385 mOsmol/kg and 7.8 pH) [32]. The semen pools in both groups were cooled in a chilling chamber at a rate of −5°C/min for 5 min, then maintained for one hour at 5°C. Each pool was then divided into three subgroups and equilibrated for a further 10 min at 5°C with a precooled DMA solution (Qualikems Fine Chem Pvt. Ltd., Vadodara, India) at a final concentration of 3%, 6%, or 9%, respectively. After that, the semen samples were uploaded into mini straws of 250 μL (IMV Technologies, L'Aigle, France). The semen was frozen by maintaining the straws horizontally at 5 cm over the liquid nitrogen ($LN_2$) surface for 10 min, then immersed in the $LN_2$. The frozen semen was thawed after 3 months of storage in the $LN_2$ tank. For semen thawing, the straws were

taken from the $LN_2$ and instantly submerged into a water bath, previously adjusted to 38°C, for 10 seconds. The thawed semen was then expelled into an Eppendorf tube and kept in the water bath for further assays.

## Detection of sperm concentration and dilution rate

Five µL of the semen pool was diluted with 995 µL of phosphate-buffered saline (PBS) containing 10% eosin (Bio-Diagnostic, Inc., Giza, Egypt). An aliquot of the dilution was pipetted onto a hemocytometer slide and scanned under a microscope at a magnification of 400×. The number of sperm in five large squares (ns) was counted to express the sperm concentration as $ns \times 10^7$ cells per mL. Immediately after determining the actual sperm concentration, the semen pool was evenly divided into two clean tubes, and a calculated volume of EK extender was added to each tube to achieve the working dilution rates for the LSC and HSC groups, respectively.

## Motility traits

Sperm motility parameters were analyzed using a computer-assisted sperm analysis (CASA; SpermVision™ software, Minitube GmbH, Tiefenbach, Germany). For reading on CASA, an aliquot (10 µL) of the post-thawed semen sample from each treatment subgroup was added to 490 µL of pre-warmed EK extender in an Eppendorf tube. After 5 min equilibration in the water bath, 6 µL of the diluted sample was placed over a prewarmed clean slide on a hotplate (37°C) and covered with a clean coverslip. A minimum of five randomly selected microscopic fields (300–500 sperm per field) were captured at 60 frames per second using a negative high-contrast microscope (Olympus-BX, Tokyo, Japan) at a total magnification of 200x. The CASA system was configured to analyze these fields. Total motile sperm (TM) percentage, progressively motile sperm (PM) percentage, average path velocity (VAP, µm/sec), curvilinear line velocity (VCL, µm/sec), straight line velocity (VSL, µm/sec), straightness (STR = VSL/VAP %), linearity (LIN = VSL/VCL %), wobble (WOB = VAP/VCL %), amplitude of lateral head displacement (ALH, µm), and beat cross frequency (BCF, Hz) were among the data evaluated by the CASA. The CASA criteria required VSL > 5 µm/sec for motile rooster sperm, and VAP > 20 µm/sec and STR > 80% for progressive motile sperm.

## Fertility traits

Ten hens were allocated for artificial insemination with the thawed semen from each treatment subgroup, following established cryopreservation protocols [33]. Using a micropipette, each hen in the HSC or LSC group received 3 intravaginal inseminations at 2-day intervals with 150 µL or 300 µL of thawed semen, respectively, containing a constant concentration of approximately $300 \times 10^6$ sperm per dose in both groups. Eggs were collected for 8 consecutive days starting from the 2nd day of initiating the insemination. The collected eggs were arranged into setter trays manufactured locally by the Poultry Technical Office (PTO, Alexandria, Egypt) and incubated for 18 days (99.6 °F temperature and 58% humidity). Then, the eggs were transferred to a hatcher (PTO) and incubated for 3 days (98.5 °F and 92 °F in dry and wet bulb readings). Upon completion of the incubation period, the quantity of hatched eggs was noted. Unhatched eggs were cracked and assorted into infertile eggs, early-dead, late-dead, and pipped embryos.

## Statistical analysis

Eight semen pools were considered to explore the data on sperm motility and fertility in the present study. Data were arranged in a completely randomized 2 × 3 factorial design of the dilution rates (LSC and HSC) and DMA concentrations (3, 6, and 9%). All statistical analysis procedures were performed using IBM SPSS Statistics 22. The effect of dilution rates, DMA concentrations, and their interactions on the sperm motility parameters was analyzed using multivariate General Linear Model (GLM) procedures. The mean values were compared using Tukey's post-hoc test. The fertility traits were analyzed using logistic regression with a binomial distribution. A polynomial contrast assay was performed to

examine the linear and quadratic effects of increasing DMA concentration on all parameters. In addition, a Pearson correlation test was done linking motility parameters with fertility outcomes. The significance level was set at $p < 0.05$.

## Results

### Sperm motility

Results showed a significant increase in the post-thaw sperm TM, VCL, ALH, and BCF, and a substantial decrease in the LIN, STR, and WOB for the dilution rate at HSC versus LSC (p<0.05) (Table 1). Regardless of dilution rates, DMA concentration significantly (p<0.05) influenced the post-thaw sperm motility parameters, except for ALH (Table 2). A notable quadratic trend was observed for DMA concentration regarding sperm TM, PM, VSL, and BCF, with the highest values

**Table 1. Effect of semen dilution rate on post-thaw motility parameters of rooster sperm.**

| Dilution rate | LSC | HSC | SEM | p-value |
|---|---|---|---|---|
| n | 126 | 133 | | |
| TM (%) | 44.4[b] | 54.7[a] | 0.96 | <0.001 |
| PM (%) | 27.5 | 27.5 | 0.66 | 0.956 |
| VCL (µm/s) | 92.9[b] | 122.0[a] | 1.63 | <0.001 |
| VSL (µm/s) | 35.4 | 35.7 | 0.46 | 0.667 |
| LIN (%) | 37.8[a] | 29.5[b] | 0.40 | <0.001 |
| STR (%) | 65.9[a] | 57.0[b] | 0.40 | <0.001 |
| WOB (%) | 57.1[a] | 51.2[b] | 0.30 | <0.001 |
| ALH (µm) | 4.4[b] | 5.4[a] | 0.05 | <0.001 |
| BCF (Hz) | 25.3[b] | 27.2[a] | 0.26 | <0.001 |

Abbreviations: LSC, dilution rate at low sperm concentration; HSC, dilution rate at high sperm concentration; TM, total motility; PM: progressive motility; VCL: velocity curved line (µm/s); VSL: velocity straight line (µm/s); STR: LIN: linearity (VSL/VCL %); straightness (VSL/VAP %); WOB: wobble (VAP/VCL %); ALH: amplitude of lateral head displacement (µm); BCF: beat cross frequency (Hz). Data are presented as means±standard error of means (SEM). Means with different superscripts, within a parameter in the same row, are significantly different (P<0.05).

**Table 2. Effect of various concentrations of dimethyl acetamide on post-thaw motility parameters of rooster sperm.**

| DMA | 3% | 6% | 9% | SEM | p-value | | |
|---|---|---|---|---|---|---|---|
| n | 89 | 84 | 86 | | DMA effect | Linear effect | Quadratic effect |
| TM (%) | 46.5[b] | 55.0[a] | 47.2[b] | 1.17 | <0.001 | 0.671 | <0.001 |
| PM (%) | 24.2[b] | 33.7[a] | 24.6[b] | 0.81 | <0.001 | 0.692 | <0.001 |
| VCL (µm/s) | 112.8[a] | 112.6[a] | 96.9[b] | 2.00 | <0.001 | <0.001 | 0.002 |
| VSL (µm/s) | 34.6[b] | 37.7[a] | 34.3[b] | 0.56 | 0.001 | 0.678 | <0.001 |
| LIN (%) | 31.7[b] | 33.9[a] | 35.3[a] | 0.50 | <0.001 | <0.001 | 0.454 |
| STR (%) | 59.8[b] | 61.3[b] | 63.3[a] | 0.50 | <0.001 | <0.001 | 0.687 |
| WOB (%) | 52.3[b] | 54.9[a] | 55.3[a] | 0.40 | <0.001 | <0.001 | 0.030 |
| ALH (µm) | 5.0 | 5.0 | 4.8 | 0.06 | 0.153 | 0.054 | 0.819 |
| BCF (Hz) | 26.1[b] | 27.4[a] | 25.1[b] | 0.31 | <0.001 | 0.025 | <0.001 |

Abbreviations: DMA, dimethyl acetamide; TM, total motility; PM: progressive motility; VCL: velocity curved line (µm/s); VSL: velocity straight line (µm/s); STR: LIN: linearity (VSL/VCL %); straightness (VSL/VAP %); WOB: wobble (VAP/VCL %); ALH: amplitude of lateral head displacement (µm); BCF: beat cross frequency (Hz). Data are presented as means±standard error of means (SEM). Means with different superscripts, within a parameter in the same row, are significantly different (P<0.05).

at 6% DMA compared to 3% or 9% DMA. Conversely, increasing DMA concentration from 3% to 9% resulted in a linear decrease (p<0.05) in VCL and a linear increase in LIN, STR, and WOB. Additionally, a significant (p<0.05) interaction effect was noted in sperm VCL, LIN, STR, WOB, and BCF (Table 3). The post-thaw sperm VCL and BCF were significantly higher in the HSC groups than LSC groups, particularly when using low DMA concentrations (3-6%). Conversely, LIN, STR, and WOB were considerably higher in the LSC groups than HSC groups, especially when using 6% DMA.

## Sperm fertility

The results of logistic regression indicated that the fertility rate was significantly (p < 0.001) higher in hens inseminated with HSC compared to the LSC, with an odds ratio (OR) of 4.05 (Table 4). However, neither DMA concentrations nor their polynomial contrasts (linear and quadratic) exerted significant effects on fertile eggs (p > 0.05) (Table 5). The interaction between dilution rate and DMA concentration on fertile eggs was not significant (Table 6). Moreover, no significant differences were observed among treatment groups regarding hatching, pipping, or embryonic mortality parameters (p > 0.05).

**Table 3. Interaction effect of semen dilution rate and dimethyl acetamide concentration on post-thaw motility parameters of rooster sperm.**

| Dilution rate | LSC | LSC | LSC | HSC | HSC | HSC | SEM | p-value |
|---|---|---|---|---|---|---|---|---|
| DMA | 3% | 6% | 9% | 3% | 6% | 9% | | |
| n | 43 | 42 | 41 | 46 | 42 | 45 | | |
| TM (%) | 40.4 | 49.5 | 43.2 | 52.5 | 60.5 | 51.1 | 1.66 | 0.418 |
| PM (%) | 23.5 | 33.8 | 25.3 | 24.9 | 33.6 | 24.0 | 1.15 | 0.465 |
| VCL (μm/s) | 93.4[c] | 98.1[bc] | 87.1[c] | 132.3[a] | 127.0[a] | 106.6[b] | 2.83 | 0.003 |
| VSL (μm/s) | 34.4 | 38.4 | 33.3 | 34.8 | 36.9 | 35.3 | 0.79 | 0.080 |
| LIN (%) | 36.6[a] | 39.0[a] | 37.8[a] | 26.7[c] | 28.9[c] | 32.9[b] | 0.67 | <0.001 |
| STR (%) | 65.4[a] | 66.8[a] | 65.4[a] | 54.1[c] | 55.7[c] | 61.3[b] | 0.78 | <0.001 |
| WOB (%) | 55.6[b] | 58.3[a] | 57.4[ab] | 49.0[d] | 51.5[c] | 53.2[c] | 0.58 | 0.033 |
| ALH (μm) | 4.5 | 4.4 | 4.4 | 5.5 | 5.5 | 5.3 | 0.08 | 0.518 |
| BCF (Hz) | 24.6[c] | 26.6[ab] | 24.6[c] | 27.7[a] | 28.2[a] | 25.6[bc] | 0.44 | 0.044 |

Abbreviations: LSC, dilution rate at low sperm concentration; HSC, dilution rate at high sperm concentration; DMA, dimethyl acetamide; TM, total motility; PM: progressive motility; VCL: velocity curved line (μm/s); VSL: velocity straight line (μm/s); STR: LIN: linearity (VSL/VCL %); straightness (VSL/VAP %); WOB: wobble (VAP/VCL %); ALH: amplitude of lateral head displacement (μm); BCF: beat cross frequency (Hz). Data are presented as means for dilution rate (LSC versus HSC), DMA concentration (3, 6, and 9%), and their interaction ± standard error of means (SEM). Means with different superscripts, within a parameter in the same row, are significantly different (P<0.05).

**Table 4. Effect of semen dilution rate on post-thaw fertility parameters of rooster sperm.**

| Dilution rate | LSC | HSC | p-value |
|---|---|---|---|
| Incubated eggs, n | 255 | 256 | |
| Fertile eggs, n (%)[1] | 17 (6.7)[b] | 58 (22.7)[a] | <0.001 |
| Hatched chicks, n (%)[2] | 11 (64.7) | 18 (31.0) | 0.999 |
| Pipped chicks, n (%)[2] | 6 (35.3) | 29 (50.0) | 0.999 |
| Early dead chicks, n (%)[2] | 0 (0.0) | 9 (15.5) | 0.998 |
| Late dead chicks, n (%)[2] | 0 (0.0) | 2 (3.4) | 0.999 |

Abbreviations: LSC, dilution rate at low sperm concentration; HSC, dilution rate at high sperm concentration; [1] Values were calculated as percentages of total incubated eggs. [2] Values were calculated as percentages of fertile eggs. Means with different superscripts, within a parameter in the same row are significantly different (P<0.05).

**Table 5. Effect of various concentrations of dimethyl acetamide on post-thaw fertility parameters of rooster sperm.**

| DMA | 3% | 6% | 9% | p-value | | |
|---|---|---|---|---|---|---|
| Incubated eggs, n | 172 | 167 | 172 | DMA effect | Linear effect | Quadratic effect |
| Fertile eggs, n (%)[1] | 29 (16.9) | 20 (12.0) | 26 (15.1) | 0.950 | 0.983 | 0.749 |
| Hatched chicks, n (%)[2] | 14 (48.3) | 10 (50.0) | 5 (19.2) | 1.000 | 0.999 | 0.999 |
| Pipped chicks, n (%)[2] | 12 (41.4) | 8 (40.0) | 15 (57.7) | 1.000 | 0.999 | 0.999 |
| Early dead chicks, n (%)[2] | 2 (6.9) | 2 (10.0) | 5 (19.2) | 1.000 | 1.000 | 1.000 |
| Late dead chicks, n (%)[2] | 1 (3.4) | 0 (0.0) | 1 (3.8) | 1.000 | 1.000 | 1.000 |

Abbreviations: DMA, dimethyl acetamide; [1] Values were calculated as percentages of total incubated eggs. [2] Values were calculated as percentages of fertile eggs. Means with different superscripts, within a parameter in the same row, are significantly different (P<0.05).

**Table 6. Interaction effect of semen dilution rate and dimethyl acetamide concentration on post-thaw fertility parameters of rooster sperm.**

| Dilution rate | LSC | LSC | LSC | HSC | HSC | HSC | p-value |
|---|---|---|---|---|---|---|---|
| DMA | 3% | 6% | 9% | 3% | 6% | 9% | |
| Incubated eggs, n | 85 | 84 | 86 | 87 | 83 | 86 | |
| Fertile eggs, n (%)[1] | 6 (7.1) | 5 (6.0) | 6 (7.0) | 23 (26.4) | 15 (18.1) | 20 (23.3) | 0.915 |
| Hatched chicks, n (%)[2] | 6 (100.0) | 5 (100.0) | 0 (0.0) | 8 (34.8) | 5 (33.3) | 5 (25.0) | 1.000 |
| Pipped chicks, n (%)[2] | 0 (0.0) | 0 (0.0) | 6 (100.0) | 12 (52.2) | 8 (53.3) | 9 (45.0) | 1.000 |
| Early dead chicks, n (%)[2] | 0 (0.0) | 0 (0.0) | 0 (0.0) | 2 (8.7) | 2 (13.3) | 5 (25.0) | 1.000 |
| Late dead chicks, n (%)[2] | 0 (0.0) | 0 (0.0) | 0 (0.0) | 1 (4.3) | 0 (0.0) | 1 (5.0) | 1.000 |

Abbreviations: LSC, dilution rate at low sperm concentration; HSC, dilution rate at high sperm concentration; DMA, dimethyl acetamide; [1] Values were calculated as percentages of total incubated eggs. [2] Values were calculated as percentages of fertile eggs. Means with different superscripts, within a parameter in the same row, are significantly different (P<0.05).

These findings indicate that the dilution rate primarily influenced post-thaw fertility outcomes, whereas DMA concentration had no consistent effect (Tables 4–6).

### Post-thaw sperm motility and fertility correlation

Pearson correlation analysis revealed that certain motility parameters were significantly associated with fertility outcomes (Table 7). VCL, ALH, and BCF showed positive correlations with fertility and pipping rates (p < 0.05), whereas STR, LIN, and WOB displayed negative correlations with fertility and pipping rates (p < 0.05). Progressive motility was negatively correlated with early death rate (p < 0.01).

## Discussion

Most poultry semen freezing protocols include a cooling phase with dilution to adjust sperm and cryoprotectant concentrations, allowing sperm acclimation to reduced temperatures [34]. Sperm concentration and dilution rate are known to influence post-thaw quality, as excessive dilution may reduce sperm longevity [35]. In the current study, two dilution techniques were compared: high (HSC; $2 \times 10^9$ sperm/mL) and low sperm concentration (LSC; $1 \times 10^9$ sperm/mL). Here, HSC significantly improved the rooster sperm TM and VCL, consistent with Iaffaldano et al. [27], Hudson et al. [36], and Gliozzi et al. [37], who reported improved sperm performance at higher concentrations, likely due to reduced cryoinjury [38]. However, our results also revealed that while HSC enhanced TM and VCL, it paradoxically reduced sperm LIN and STR, with no significant effects on PM and VSL. Similar trade-offs have been reported in avian species, where increased sperm concentration elevates motility through sperm–sperm interactions and hyperactivation-like movement, but frequent

**Table 7. Pearson correlation linking motility parameters with fertility outcomes of post-thaw rooster sperm.**

|  | Fertility | Hatchability | Pipping eggs | EDR | LDR |
|---|---|---|---|---|---|
| **TM** | 0.041 | 0.245 | 0.133 | −0.161 | −0.077 |
| **Prog** | −0.142 | 0.224 | −0.149 | −0.332** | −0.052 |
| **VCL** | 0.287* | 0.258* | 0.343** | 0.025 | −0.016 |
| **VSL** | −0.051 | 0.191 | −0.024 | −0.064 | −0.020 |
| **STR** | −0.313* | −0.130 | −0.447** | −0.094 | 0.033 |
| **LIN** | −0.327* | −0.152 | −0.414** | −0.086 | 0.017 |
| **WOB** | −0.332** | −0.182 | −0.323* | −0.065 | −0.014 |
| **ALH** | 0.284* | 0.110 | 0.501** | 0.169 | −0.021 |
| **BCF** | 0.210 | 0.141 | 0.317* | 0.029 | −0.018 |

Abbreviations: TM, total motility; PM: progressive motility; VCL: velocity curved line (µm/s); VSL: velocity straight line (µm/s); STR: LIN: linearity (VSL/VCL %); straightness (VSL/VAP %); WOB: wobble (VAP/VCL %); ALH: amplitude of lateral head displacement (µm); BCF: beat cross frequency (Hz); EDR, early death rate; LDR, late death rate. The data represent the values of the correlation coefficients. The sign of the coefficient indicates the direction of the correlation, and the absolute value indicates the strength of the correlation. *Correlation is significant at p < 0.05. **Correlation is significant at p < 0.01.

collisions and hydrodynamic interference reduce directional movement [39–41]. Thus, the biological balance observed—higher motility but lower trajectory efficiency—highlights how sperm concentration shapes post-thaw kinematics and may critically influence fertilization outcomes in poultry.

On the other hand, the present study revealed that fertile eggs in the HSC group were at least 4-fold higher than those in the LSC group, likely due to superior motility indices (TM, VCL, and ALH) that help preserve flagellar activity and fertilizing capacity [42,43]. Excessive dilution, as in LSC group, may reduce seminal plasma antioxidants and proteins essential for membrane stability and sperm function, impairing the motility and fertilization ability [26,44–46]. However, despite improved fertilization in HSC group, hatchability of LSC-fertile eggs seemed to be higher (p>0.05), possibly due to polyspermy, or higher embryonic mortality in over-concentrated groups [47–49]. These findings highlight the delicate balance between sperm concentration and reproductive outcomes.

With respect to DMA effects, our findings revealed a significant influence on sperm motility characteristics but not on fertility outcomes. These results indicate that DMA primarily modulates sperm kinematics rather than fertilizing ability, aligning with earlier reports that cryoprotectants impact motility more consistently than reproductive performance [6,50,51]. Regardless of dilution rates, DMA concentration significantly (p < 0.05) affected most motility parameters except ALH, with quadratic trends for TM, PM, VSL, and BCF—peaking at 6% DMA (Table 2). Interaction effects between dilution rate and DMA were also evident, with HSC groups generally exhibiting higher VCL and BCF, while LSC groups showed superior LIN, STR, and WOB, particularly at 6% DMA (Table 3). These results support earlier studies indicating that cryoprotectants, including DMA, have a concentration-dependent impact on the post-thaw *in vitro* quality of cryopreserved chicken sperm [5,51] and, when used at moderate concentrations, they balance sperm membrane protection and cytotoxicity [52]. Higher DMA concentrations, as 9% in the present study, appeared detrimental, possibly due to toxic effects on sperm membranes and DNA integrity [53]. It could be noticed that DMA at 9% deteriorated sperm fertility traits, although not significantly, recording lower hatchability rates and higher pipping chicks of fertile eggs than DMA at 3–6% (Table 5). Such negative effects could be due to the sensitivity of chicken spermatozoa to the small-molecular-weight intracellular cryoprotectants, such as DMA, which induce subtle damage to the sperm membrane and disturb the antioxidant defence system [51,54]. Moreover, using DMA at 9% in the semen freezing protocols may downregulate the expression of some genes critically associated with sperm biological processes, such as heat shock protein 70 (*HSP70*) and ras homolog family member A (*RHOA*) [6]. These events may also explain

the low fertilization rates, ranging from 6.0 to 26.4%, after inseminating the hens with the post-thaw frozen semen in the present study.

CASA is the routine laboratory tool for kinematic profiling, but values depend on several factors, including smoothing/threshold settings, temperature, pH, dilution/extender, cryoprotectants, and freeze/thaw procedures [6,7,55]. The VCL and VSL indicate vigorous and progressive movement essential for oviductal transit, the LIN and STR reflect the efficiency and directionality of sperm trajectories, and the WOB, ALH, and BCF are associated with flagellar activity and hyperactivation-like motility, facilitating perivitelline penetration [16,56]. Practically, CASA kinematics after cryopreservation are essential for male donor selection, cryoprotectant choice, and insemination dose validation, thereby contributing to more efficient cryopreservation protocols and biobanking programs [57]. However, it has been suggested that sperm motility parameters may not fully predict fertility after freeze-thaw in poultry species due to sperm cryosensitivity [6]. In addition, PM, VAP, VSL, and VCL may strongly present inter-correlation to each other, but fertility is a multifactorial process that involves female effects, insemination timing and dose, and structural and functional integrity of sperm cells rather than involving sperm movement characteristics [58,59]. In the present study, the correlation analysis revealed that VCL, LIN, STR, WOB, ALH, and BCF may serve as more accurate predictors of post-thaw fertility outcomes than conventional TM or PM (Table 7). Similarly, Farooq et al. [60] concluded that CASA sperm motility analysis is partially successful for the estimation of fertility in Japanese quail, indicating that PM is important for male evaluation, while VCL, VAP, and BCF are important sperm traits for predicting egg fertility. In contrast, Muvhali et al. [41] reported that mass sperm motility in farmed ostrich correlated positively with TM, PM, VCL and VAP but negatively with LIN, STR, and BCF, showing that different metrics can capture distinct aspects of sperm performance relevant to fertility. Therefore, the discrepancies observed in correlation results may reflect the complexity of fertilization, where multiple semen quality traits act in concert rather than motility functioning as an independent determinant.

## Conclusions

Our findings suggest that applying high sperm concentration and 6% DMA for chicken semen freezing may best maintain the post-thaw sperm motility and fertility outcomes after artificial insemination. Further studies evaluating sperm functional integrity (e.g., acrosome status, DNA fragmentation) and *in vivo* fertility trials are warranted to understand the mechanism of action and refine these protocols.

## Supporting information

**S1 File. Data set 1.**
(PDF)

**S2 File. Data set 2.**
(PDF)

## Author contributions

**Conceptualization:** Mohamed Shawky, Mohamed A. Elmenawey, Abd El-Rahman M. Atta, Ahmed O. Abbas, Gamal M. K. Mehaisen.

**Data curation:** Ahmed F. Fawy, Ahmed M. Elomda, Abd El-Rahman M. Atta, Gamal M. K. Mehaisen.

**Formal analysis:** Mohamed Shawky, Mohamed A. Elmenawey, Ahmed O. Abbas, Gamal M. K. Mehaisen.

**Funding acquisition:** Mohamed Shawky, Gamal M. K. Mehaisen.

**Investigation:** Mohamed Shawky, Ahmed F. Fawy, Ahmed M. Elomda, Mohamed A. Elmenawey, Abd El-Rahman M. Atta, Ahmed O. Abbas, Gamal M. K. Mehaisen.

**Methodology:** Ahmed F. Fawy, Ahmed M. Elomda, Mohamed A. Elmenawey, Abd El-Rahman M. Atta, Gamal M. K. Mehaisen.

**Project administration:** Mohamed Shawky, Gamal M. K. Mehaisen.

**Resources:** Mohamed Shawky.

**Supervision:** Mohamed A. Elmenawey, Abd El-Rahman M. Atta, Ahmed O. Abbas, Gamal M. K. Mehaisen.

**Validation:** Ahmed M. Elomda, Mohamed A. Elmenawey.

**Writing – original draft:** Ahmed F. Fawy, Ahmed M. Elomda, Ahmed O. Abbas, Gamal M. K. Mehaisen.

**Writing – review & editing:** Mohamed Shawky, Ahmed F. Fawy, Ahmed M. Elomda, Mohamed A. Elmenawey, Abd El-Rahman M. Atta, Ahmed O. Abbas, Gamal M. K. Mehaisen.

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
