## [Decision Letter · Decision Letter 0]

27 Aug 2025

PONE-D-25-33278
Effect of pre-freezing semen dilution rate and cryo-protection with dimethyl acetamide on post-thaw motility and fertility traits of rooster sperm
PLOS ONE

Dear Dr. Gamal M. K. Mehaisen

Thank you for submitting your manuscript to PLOS ONE. After careful consideration, we feel that it has merit but does not fully meet PLOS ONE’s publication criteria as it currently stands. Therefore, we invite you to submit a revised version of the manuscript that addresses the points raised during the review process.

We look forward to receiving your revised manuscript.

Kind regards,

Sameh Abdelnour

Academic Editor

PLOS ONE

Journal Requirements:

2. To comply with PLOS One submissions requirements, in your Methods section, please provide additional information regarding the experiments involving animals and ensure you have included details on (1) methods of sacrifice, and efforts to alleviate suffering.

“This research was financially supported by the funds from the Academy of Scientific Research and Technology, Egypt (ASRT 5105) and the Vice Presidency for Graduate Studies and Scientific Research at King Faisal University, Saudi Arabia (Project KFU252322).”

 “This research was financially supported by the funds from the Academy of Scientific Research and Technology, Egypt (ASRT 5105) and the Vice Presidency for Graduate Studies and Scientific Research at King Faisal University, Saudi Arabia (Project KFU252322).”

5. We note that your Data Availability Statement is currently as follows: [All relevant data are within the manuscript and its Supporting Information files.]

6. Thank you for stating the following in your manuscript:

“This research was financially supported by the funds from the Academy of Scientific Research and Technology, Egypt (ASRT 5105) and the Vice Presidency for Graduate Studies and Scientific Research at King Faisal University, Saudi Arabia (Project KFU252322).”

“This research was financially supported by the funds from the Academy of Scientific Research and Technology, Egypt (ASRT 5105) and the Vice Presidency for Graduate Studies and Scientific Research at King Faisal University, Saudi Arabia (Project KFU252322).”

Reviewers' comments:

Reviewer's Responses to Questions

**Comments to the Author**

1. Is the manuscript technically sound, and do the data support the conclusions?

Reviewer #1: Yes

Reviewer #2: Partly

2. Has the statistical analysis been performed appropriately and rigorously? 

Reviewer #1: Yes

Reviewer #2: No

3. Have the authors made all data underlying the findings in their manuscript fully available?

Reviewer #1: Yes

Reviewer #2: No

4. Is the manuscript presented in an intelligible fashion and written in standard English?

Reviewer #1: Yes

Reviewer #2: No

5. Review Comments to the Author

Reviewer #1: The manuscript presents a well-designed study addressing an important topic in poultry reproduction. The experimental design is robust, data analysis is appropriate, and conclusions are supported by the results. Minor revisions are required to clarify methodological details and improve data presentation.

1. Check the chart format, the indicators and values do not correspond; Split table presentation is recommended; It is recommended to clarify the number of Hatched chicks, etc;

2. It is recommended to supplement other important experimental results in addition to data, such as pictures;

3. The overall fertilization rate result is low, please analyze and discuss this result;

4. The format of the references does not agree to be unified and there are format errors, please correct them; It is recommended to refer to literature from recent years.

Reviewer #2: Title

• Please consider making the title more precise, for example: “Effect of semen dilution rate and dimethyl acetamide levels on post-thaw motility and fertility parameters of rooster sperm.”

Abstract

• The concluding sentence needs improvement. It should be more practically oriented, emphasizing potential applications in field conditions.

Introduction

• Please highlight the novelty of the study more explicitly.

• Line 49: The phrase “freezing technique per se” is unclear and requires clarification.

Materials and Methods

• Line 113: Specify the initial temperature at the time of dilution.

• Lines 115–118: Provide the source and catalog numbers for all products used.

• Line 119: Clarify the cooling rate, i.e., the time required for lowering the temperature from the initial to the final (5 °C).

• Line 126: Verify and clearly state the temperature and corresponding time duration.

• Line 138: The statement “After dilution with pre-warmed EK extender to a final ratio of 1:50 (v/v)” is unclear. Please rephrase for clarity.

• Line 153: The insemination dose (300 × 10⁶ sperm) requires a reference to justify its selection.

• Line 154: Specify the volume of thawed semen used for each insemination.

Statistical Analysis

• The fertility data is not appropriate for Chi-Square analysis. Logistic regression with a suitable distribution (e.g., Binomial, events/trials) is recommended.

• What was the experimental unit for the fertility data: eggs or hens? Apparently, it looks like eggs, which is not correct. Also, provide the main effects of dose (Low or High concentrations of sperm) or DMA concentrations and interaction, along with polynomial contrast with fertility data.

• Indicate the experimental design used (e.g., split-plot or completely randomized design).

Results

• Lines 180–181: This statement is a repetition of the previous one and should be removed.

• Tables: Place the p-values of main effects, interactions, and contrasts in columns instead of rows, it’ll improve readability.

• Table 1: Please confirm whether “n” for LSC and HSC groups is 54 or 72.

• Table 2: Provide both numerical values and percentages (e.g., for fertile eggs under LSC treatment, report “n” alongside 6.7%).

• Include a correlation table linking motility parameters with fertility outcomes. Try to establish which motility parameters more accurately describe the fertility outcomes.

Discussion

• The discussion requires substantial improvement; it is too lengthy and at times repetitive.

• Address discrepancies observed in the results (e.g., why the HSC resulted in increased early and late chick mortality).

• Elaborate the practical importance of sperm motility parameters (VCL, VSL, LIN, STR, WOB, ALH, BCF) and their relevance for field application.

6. PLOS authors have the option to publish the peer review history of their article (what does this mean?). If published, this will include your full peer review and any attached files.

Reviewer #1: No

Reviewer #2: No

---

## [Author Response · Author response to Decision Letter 1]

8 Oct 2025

Response to Reviewer #1

The manuscript presents a well-designed study addressing an important topic in poultry reproduction. The experimental design is robust, data analysis is appropriate, and conclusions are supported by the results. Minor revisions are required to clarify methodological details and improve data presentation.

1. Check the chart format, the indicators and values do not correspond; Split table presentation is recommended; It is recommended to clarify the number of Hatched chicks, etc;

Response:

We represented the tables after some modifications requested by the reviewers.

2. It is recommended to supplement other important experimental results in addition to data, such as pictures.

Response:

Unfortunately, we did not have appropriate pictures to present in the manuscript. We hope modifications applied to the tables upon reviewers' requests are currently sufficient.

3. The overall fertilization rate result is low, please analyze and discuss this result.

Response:

We added a phrase explaining this result in the discussion (Lines 453-471).

4. The format of the references does not agree to be unified and there are format errors, please correct them; It is recommended to refer to literature from recent years..

Response:

We used EndNote Reference Manager to insert the citations and reference list in the manuscript. We manually revised the reference list one by one to eliminate such errors.

Response to Reviewer #2

Title

• Please consider making the title more precise, for example: “Effect of semen dilution rate and dimethyl acetamide levels on post-thaw motility and fertility parameters of rooster sperm.”

Response:

The recommended title was considered (Line 4-6).

Abstract

• The concluding sentence needs improvement. It should be more practically oriented, emphasizing potential applications in field conditions.

Response:

It was improved (Lines 44-49).

Introduction

• Please highlight the novelty of the study more explicitly.

Response:

The final paragraph was improved to highlight the study's objective and novelty (Lines 105-115).

• Line 49: The phrase “freezing technique per se” is unclear and requires clarification.

Response:

The phrase was clarified (Lines 55-62).

Materials and Methods

• Line 113: Specify the initial temperature at the time of dilution.

Response:

It was specified (Line 145).

• Lines 115–118: Provide the source and catalog numbers for all products used.

Response:

It was provided (Lines 146-151).

• Line 119: Clarify the cooling rate, i.e., the time required for lowering the temperature from the initial to the final (5 °C).

Response:

It was clarified (Line 153).

• Line 126: Verify and clearly state the temperature and corresponding time duration.

Response:

It is a practical procedure for quickly thawing the straws. The method is straightforward: take the straw from liquid nitrogen, immediately place it in a water bath previously adjusted to 38 °C, and keep it inside for 10 seconds. This will thaw the semen in the straw and make it ready for further assays. We have slightly improved this sentence to make it clearer for the reader (Lines 160-162).

• Line 138: The statement “After dilution with pre-warmed EK extender to a final ratio of 1:50 (v/v)” is unclear. Please rephrase for clarity.

Response:

It was clarified (Line 175-179).

• Line 153: The insemination dose (300 × 10⁶ sperm) requires a reference to justify its selection.

Response:

It was provided (Line 194).

• Line 154: Specify the volume of thawed semen used for each insemination.

Response:

As we unify the sperm concentration for insemination of hens in both groups (LSC and HSC), the volume of inseminated semen differs by group. It was specified in the text (Lines 194-197).

Statistical Analysis

• The fertility data is not appropriate for Chi-Square analysis. Logistic regression with a suitable distribution (e.g., Binomial, events/trials) is recommended.

Response:

The data were reanalyzed using logistic regression, and the statistical analysis section was updated (Line 214-215).

• What was the experimental unit for the fertility data: eggs or hens? Apparently, it looks like eggs, which is not correct. Also, provide the main effects of dose (Low or High concentrations of sperm) or DMA concentrations and interaction, along with polynomial contrast with fertility data.

Response:

The data were reanalyzed using logistic regression, and the results are presented in Tables 4-6.

• Indicate the experimental design used (e.g., split-plot or completely randomized design).

Response:

The study employed a completely randomized design (Lines 208-210).

Results

• Lines 180–181: This statement is a repetition of the previous one and should be removed.

Response:

It was deleted from lines 227-229.

• Tables: Place the p-values of main effects, interactions, and contrasts in columns instead of rows, it’ll improve readability.

Response:

OK, it was done upon your request. However, to maintain harmony and clarity in the results presentation, we divided each of Tables 1 and 2 into three separate tables, representing the results of the dilution rate, DMA, and their interactions, respectively (Tables 1-6). Based on this change, the CASA and fertility results were rewritten (Lines 225-240 and 280-288).

• Table 1: Please confirm whether “n” for LSC and HSC groups is 54 or 72.

Response:

Thank you very much for your notice. We carefully revised the current statistical analysis and corrected these mistakes. We updated the tables to make them more comfortable and readable. Please refer to Tables 1-3 for CASA results and Tables 4-6 for fertility outputs.

• Table 2: Provide both numerical values and percentages (e.g., for fertile eggs under LSC treatment, report “n” alongside 6.7%).

Response:

OK, it was done upon your request (Lines 312-329).

• Include a correlation table linking motility parameters with fertility outcomes. Try to establish which motility parameters more accurately describe the fertility outcomes.

Response:

OK, it was done upon your request. We added new parts to the statistical analysis (Lines 218-219) and the results (Lines 331-346 and Table 7).

Discussion

• The discussion requires substantial improvement; it is too lengthy and at times repetitive.

Response:

The discussion was carefully revised and improved.

• Address discrepancies observed in the results (e.g., why the HSC resulted in increased early and late chick mortality).

Response:

The observed discrepancies were discussed (e.g., Lines 412-418 and 490-506).

• Elaborate the practical importance of sperm motility parameters (VCL, VSL, LIN, STR, WOB, ALH, BCF) and their relevance for field application.

Response:

It was elaborated and discussed (Lines 482-490).

---

## [Editor Report · Decision Letter 1]

15 Oct 2025

Effect of semen dilution rate and dimethyl acetamide levels on post-thaw motility and fertility parameters of rooster sperm

PONE-D-25-33278R1

Dear Dr. Gamal M. K. Mehaisen,

We’re pleased to inform you that your manuscript has been judged scientifically suitable for publication and will be formally accepted for publication once it meets all outstanding technical requirements.

Kind regards,

Sameh Abdelnour

Academic Editor

PLOS ONE
---

## [Editor Report · Acceptance letter]

PONE-D-25-33278R1

PLOS ONE

Dear Dr. Mehaisen,

I'm pleased to inform you that your manuscript has been deemed suitable for publication in PLOS ONE. Congratulations! Your manuscript is now being handed over to our production team.

Kind regards,

on behalf of

Dr. Sameh Abdelnour

Academic Editor

PLOS ONE